# Molecular and Cytogenetic Analysis of Romanian Patients with Differences in Sex Development

**DOI:** 10.3390/diagnostics11112107

**Published:** 2021-11-14

**Authors:** Diana Miclea, Camelia Alkhzouz, Simona Bucerzan, Paula Grigorescu-Sido, Radu Anghel Popp, Ionela Maria Pascanu, Victoria Cret, Cristina Ghervan, Ligia Blaga, Gabriela Zaharie

**Affiliations:** 1Molecular Sciences Department, “Iuliu Hatieganu” University of Medicine and Pharmacy, 400012 Cluj-Napoca, Romania; anghel.popp@umfcluj.ro; 2Medical Genetics Department, Clinical Emergency Hospital for Children, 400370 Cluj-Napoca, Romania; victoria_cret@yahoo.com; 3Mother and Child Department, “Iuliu Hatieganu” University of Medicine and Pharmacy, 400012 Cluj-Napoca, Romania; p_grigorescusido@yahoo.com (P.G.-S.); blagaligia@yahoo.com (L.B.); gabrielazaharie1966@gmail.com (G.Z.); 4Endocrinology Department, George Emil Palade University of Medicine, Pharmacy, Science and Technology, 540142 Targu Mures, Romania; iopascanu@gmail.com; 5Endocrinology Department, “Iuliu Hatieganu” University of Medicine and Pharmacy, 400012 Cluj-Napoca, Romania; cghervan@yahoo.com

**Keywords:** differences in sex development, karyotype, SRY, CYP21A2, chromosomal microarray, next-generation sequencing

## Abstract

Differences in sex development (DSD) are often correlated with a genetic etiology. This study aimed to assess the etiology of DSD patients following a protocol of genetic testing. Materials and methods. This study prospectively investigated a total of 267 patients with DSD who presented to Clinical Emergency Hospital for Children Cluj-Napoca between January 2012 and December 2019. Each patient was clinically, biochemically, and morphologically evaluated. As a first intervention, the genetic test included karyotype + *SRY* testing. A high value of 17-hydroxyprogesterone was found in 39 patients, in whom strip assay analysis of the *CYP21A2* gene was subsequently performed. A total of 35 patients were evaluated by chromosomal microarray technique, and 22 patients were evaluated by the NGS of a gene panel. Results. The karyotype analysis established the diagnosis in 15% of the patients, most of whom presented with sex chromosome abnormalities. Genetic testing of *CYP21A2* established a confirmation of the diagnosis in 44% of patients tested. SNP array analysis was particularly useful in patients with syndromic DSD; 20% of patients tested presented with pathogenic CNVs or uniparental disomy. Gene panel sequencing established the diagnosis in 11 of the 22 tested patients (50%), and the androgen receptor gene was most often involved in these patients. The genes that presented as pathogenic or likely pathogenic variants or variants of uncertain significance were *RSPO1*, *FGFR1*, *WT1*, *CHD7*, *AR*, *NIPBL*, *AMHR2*, *AR*, *EMX2*, *CYP17A1*, *NR0B1*, *GNRHR*, *GATA4*, and *ATM* genes. Conclusion. An evaluation following a genetic testing protocol that included karyotype and *SRY* gene testing, *CYP21A2* analysis, chromosomal analysis by microarray, and high-throughput sequencing were useful in establishing the diagnosis, with a spectrum of diagnostic yield depending on the technique (between 15 and 50%). Additionally, new genetic variants not previously described in DSD were observed.

## 1. Background

Differences in sex development (DSD), defined by atypical developments of chromosomal, gonadal, or phenotypic sex, are observed in 1:4000 newborns, although this incidence may differ according to the disorders included. Isolated hypospadias or cryptorchidism are more frequently observed, in as many as 1:200 newborns [1,2,3,4]. These disorders are at the end of the DSD spectrum, and in some cases are not included as DSD, even if in some cases, they represent a milder effect of the same etiology seen in severe DSD.

Clinical features that indicate DSD usually include isolated clitoral hypertrophy, isolated posterior hypospadias, bilateral cryptorchidism or ectopia, and unilateral cryptorchidism/testicular ectopia associated with hypospadias or micro-penis; these signs are often seen in the newborn period [5,6]. Signs that suggest DSD at puberty are virilization of initially female genitalia, pubertal delay, and primary amenorrhea [5,6]. Usually, the clinical investigation aims to examine the following: genital tubercle (size and aspect), labioscrotal folds (their posterior fusion), the urinary meatus (localization), and palpable gonads. Based on these data, the Prader stages—a masculinization scale of the external genitalia and anogenital distance—could be useful indices for the degree of genital virilization [5,6,7]. Clinical evaluation of the external genitalia is always followed by an imaging evaluation to describe the characteristics of the urogenital sinus, Mullerian structures, gonads, adrenal glands, and sometimes, the renourinary system (to determine the common embryonic origin). This evaluation usually is performed using ultrasounds and MRI, although genitography/genitoscopy or even exploratory laparoscopy and histopathological examination are utilized in some situations [8].

DSD are classified depending on karyotype (i.e., 46,XY DSD; 46,XX DSD; and sex chromosome DSD), and in some cases, DSD is observed as part of other developmental disorders (e.g., syndromic DSD) [9]. 46,XY and 46,XX DSD could be the results of abnormal gonadal development or disorders of steroid synthesis and action. 46,XX disorders of ovarian development include 46,XX testicular and ovotesticular DSD, which are caused by *SRY* translocation in 85% of testicular 46,XX and 10% of ovotesticular 46,XX, as well as *SOX9*, *RSPO1*, or *SOX3* disorders. They also include 46,XX gonadal dysgenesis, which, in cases of premature ovarian failure, is due to genes involved in gonadal development, such as *FSHR*, *NR5A1*, *BMP15*, *STAG3*, and *WT1* [9,10]. 46,XY disorders of testicular development include complete and partial gonadal dysgenesis, which are due more often to *SRY* gene abnormalities and less often to *DHH*, *NR5A1*, *MAP3K1*, *CBX2*, *NROB1*, *WNT4*, or *DMRT1* gene abnormalities [9,10]. Sex chromosome DSD includes Turner syndrome (45,X and variants); Klinefelter syndrome (47,XXY and variants); mosaic 45,X/46,XY (mixed gonadal dysgenesis); 46,XX/46,XY (ovotesticular DSD); and other aneuploidies involving sex chromosomes [9]. 

DSD may not be an endocrine disorder but rather a developmental one, such as cloacal exstrophy (omphalocele-exstrophy-imperforate anus-spinal defects–OEIS spectrum (in 46,XX and 46,XY DSD) and developmental abnormalities of the Mullerian structures (e.g., hypospadias in 46,XY DSD or Mayer-Rokitansky-Küster-Hauser (MRKH) syndrome, vaginal atresia, and Mullerian agenesis in 46,XX DSD). In cases of 46,XX and 46,XY disorders of steroid hormone biosynthesis, *CYP21A2* gene mutations are the most common cause (>90% of these cases). Genetic mutations found among the remainder include: *STAR* (46,XY DSD), *3-βHSD* (*46*,*XY DSD*), *CYP11A1* (*46*,*XY DSD*), *CYP11B1* (*46*,*XX DSD*), *POR* (46,XX and 46,XY DSD), *CYP17A1* (46,XY DSD), *CYP19A1* (*46*,*XX DSD*), *5ARD* (*46*,*XY DSD*), *or 17βHSD3* (*46*,*XY DSD*) [9]. Disorders of steroid hormone action in 46,XY patients are most often due to gene mutations in *AR*, *AMHR*, or *LHR* abnormalities.

Morphologic assessment is always based on biochemical and genetic analysis. Biochemical analysis refers primarily to steroid analysis, such as plasma 17-hydroxyprogesterone, testosterone, dihydrotestosterone, androstenedione, DHEA, DHEAs, deoxycorticosterone, plasma renin activity, ACTH, FSH, LH, AMH, and urinary/24-h evaluation of urinary steroid profile (performed by gas chromatography associated with mass spectrometry [GC/MS]) [11]. Plasma sexual steroids are examined with immunoassay or liquid chromatography associated with tandem mass spectrometry (LC-MS/MS) [11].

Genetic testing is based on first-tier testing, as well as karyotype analysis and *SRY* gene evaluation, which aims to establish the first step in understanding the pathogenetic mechanism [9,12]. Then, depending on the clinical and hormonal picture, the *CYP21A2* gene is assessed by Sanger sequencing if 17-hydroxyprogesterone indicates an adrenal enzymatic block. If not, a gene panel or exome, or whole genome will be performed using next-generation sequencing (NGS) techniques [13]. If DSD is associated with other symptoms (e.g., syndromic DSD), chromosomal analysis by microarray is performed to observe the copy number variants (CNVs) that may have caused the disorder. Knowing the precise diagnosis will help in understanding the prognosis and designing better treatments that consider a precise etiopathogenetic mechanism. Therefore, the aim of this study was to assess the genetic etiology of patients with DSD presented in our service, who were clinically, biochemically, and anatomically evaluated.

## 2. Materials and Methods

Patients with DSD who presented to Clinical Emergency Hospital for Children Cluj-Napoca between January 2012 and December 2019 were investigated prospectively. The inclusion criteria for the study group were clitoral hypertrophy, posterior hypospadias, bilateral cryptorchidism or ectopia, unilateral cryptorchidism/testicular ectopia associated with hypospadias or micro-penis, puberty delay, and primary amenorrhea. A total of 267 patients were evaluated (Figure 1) with karyotype and *SRY* testing (using either fluorescent in situ hybridization (FISH) or polymerase chain reaction (PCR) techniques).

Patients were evaluated by imaging studies (ultrasound, pelvic MRI), with additional studies (such as exploratory laparotomy and gonadal biopsy) performed depending on the clinical context. Hormonal testing (for 17-hydroxyprogesterone, DHEAS, delta4 androstenedione, testosterone, DHT, and AMH) was also performed for some patients depending on the clinical context. 

In 46,XX patients with high 17-hydroxyprogesterone values (greater than 2 ng/mL), testing for 21-hydroxylase deficiency was performed. For some patients, genetic testing was performed if 17-hydroxyprogesterone was over 10 ng/mL after a stimulation test with synthetic ACTH. Forty-one 46,XX patients were genetically tested using strip assay analysis for the 11 most common mutations of *CYP21A2* (Figure 1).

For patients for whom karyotype, SRY, and 21-hydroxylase deficiency testing did not establish a diagnosis, the SNP array technique was used to evaluate CNVs (copy number variants) if the patients continued to be evaluated. Some patients declined follow-up investigations precluding obtaining further clinical data or biological samples. These patients had no clinical, hormonal, or morphological differences from the group further tested. Therefore, given limited financial funds for genomic testing (SNP array or high throughput sequencing) provided by our healthcare system, it was decided to perform these tests only for the patients who continued the follow-up for their disorders. 

SNP array testing was completed for 35 patients (Figure 1), of whom 22 patients had negative results and were consequently evaluated with gene panel sequencing (TruSight One panel, Illumina) (Figure 1). This panel included known genes associated with human pathology, particularly those associated with DSD. 

The research was approved by the ethics committee of “Iuliu Hatieganu” University of Medicine and Pharmacy of Cluj-Napoca. Written informed consent was obtained from the parents of all patients in the study.

### 2.1. Strip Assay Technique for the CYP21A2 Gene

The DNA from 3 mL of peripheral blood from each patient was purified using a Wizard^®^ Genomic DNA Purification Kit (Promega, Madison, WI, USA). This was performed using a PCR amplification followed by hybridization on a strip containing specific oligonucleotide probes for the 11 most common mutations of the *CYP21A2* gene: P30L, I2 splice (I2 G), Del 8 bp E3 (G110del8nt), I172N, Cluster E6 (I236N, V237E, M239K), V281L, L307 frameshift (F306+T), Q318X, R356W, P453S, and R483P (^®^ViennaLab CAH Strip Assay, Vienna, Austria) [14].

### 2.2. Chromosomal Microarray Technique

The chromosomal microarray comprised an SNP array, which was performed using an Infinium OmniExpress-24 BeadChip array kit (Illumina, San Diego, CA, USA), and the platform iScan System (Illumina, San Diego, CA, USA). The bioinformatic instrument was Genome Studio software version 2.0 (Illumina, San Diego, CA, USA). This analysis was based on 700,000 markers, and the interpretation of the results was based on the American College of Medical Genetics (ACMG) recommendation [15]. 

### 2.3. Genes Panel Sequencing

The sequencing was performed using the TruSight One Kit (Illumina, San Diego, CA, USA), which targets 4800 genes associated with human pathology (12 Mb). This included around 150 genes or candidate genes associated with the clinical phenotype for DSD. The sequencing was performed with the MiSeq platform (Illumina, San Diego, CA, USA) [16] using the manufacturer’s instructions. Bioinformatic analysis was performed using Galaxy bioinformatic platform, and variant interpretation was based on ACMG recommendations [17].

## 3. Results

A total of 267 patients were evaluated with karyotype and *SRY* testing. Of these patients, 39 (14.6%) were diagnosed with different chromosomal abnormalities, most of them involving the sex chromosomes (36 of 39 patients; Table 1). The 46,XX karyotype was observed in 104 patients (39%), while the 46,XY karyotype was observed in 122 patients (46%). Two patients with the 46,XX karyotype were *SRY* positive. 

Of patients with the 46,XX karyotype, 39 were tested for the main *CYP21A2* mutations (using strip assay). This testing was performed if their basal 17-hydroxyprogesterone was greater than 2 ng/mL or if stimulated 17-hydroxyprogesterone (after synthetic ACTH administration) was greater than 10 ng/mL. In 17 (44%) of the 39 patients, the diagnosis was confirmed.

Of 85 patients with the 46,XX karyotype (without an established etiologic diagnosis, after *CYP21A2* testing) and of 122 patients with the 46,XY karyotype, 8 and 27 patients, respectively, were further analyzed using chromosomal microarray (Table 2). The other undiagnosed patients were not tested further for different reasons (i.e., absent biological sample, absent clinical follow-up, limited possibility of genomic testing). Three of the eight 46,XX patients (38%) presented pathogenic CNVs (two patients) or VUS (variant of unknown significance)(one patient); two of these patients presented syndromic 46,XX DSD, while one presented an isolated form (Table 2).

Of the 27 46,XY patients, 10 patients (37%) were diagnosed with pathogenic CNVs (three patients), uniparental disomy (three patients), or VUS (four patients); each of these 10 patients presented syndromic DSD. 

Patient p10, 46,XY, presented a recurrent 16p11.2 deletion of 597kb, which has classically been associated with intellectual disability, developmental delay, autism spectrum disorders, and obesity. Cryptorchidism has also been mentioned in patients with this CNV. However, the precise gene responsible for cryptorchidism is not known. Patient p4 presented with disomy of chromosome 7, and his phenotype was concordant with the Russell-Silver phenotype. This syndrome has been associated with cryptorchidism, micro-penis, and hypospadias in other patients, and thus the genital phenotype in this patient was considered a result of this syndrome. Two patients, p14 and p21, presented 15 chromosome isodisomy, and further MLPA identified a maternal origin for these chromosomes. The clinical phenotype superposed that of Prader-Willi syndrome, and in this clinical context, the cryptorchidism was due to hypogonadotropic hypogonadism. Patient p18, who had an exon 4 deletion in the *OTC* gene, presented cryptorchidism. *OTC* mutations have not been previously described in association with this phenotype; however, the metabolic defect associated with hypotonia at a younger age may have had some association with cryptorchidism. Patients p3 and p6 presented uncertain CNVs in the 15q11.2 region, but previous data have not supported the involvement of these CNVs in DSD. Patients p24, p26, and p35 presented variants of uncertain significance, with no clear argument for their consideration in a final diagnosis. 

P30, a 46,XX patient, presented 1q21.1 deletion, or BP2-BP3 (thrombocytopenia absent radius [TAR] syndrome), and the clinical phenotype of the patient superposed with this syndrome. Genitourinary abnormalities and Mayer-Rokitansky-Küster-Hauser syndrome have been described in TAR syndrome, and this diagnosis was etiologically appropriate for the uterus agenesia described in patient p30. The gene *RBM8A* is suggested to be responsible for the Mayer-Rokitansky-Küster-Hauser phenotype in TAR syndrome. 

Patient p31 was diagnosed with 18p11.32-18p11.31 deletion, an etiological diagnosis for genitourinary abnormality, cryptorchidism, and micro-penis that have been described in other patients with this deletion. The *CYP21A2* homozygous deletion in 46,XX patient p32 identified by SNP array technique was also an etiologic diagnosis for this patient. Chromosomal microarray led to a clear etiologic diagnosis for two of eight 46,XX patients (25%) and for five of 27 46,XY patients (19%). Thus, of the 35 patients tested (46,XX and 46,XY), 7 (20%) had a final etiologic diagnosis. 

Patients with negative results after SNP array testing were analyzed by gene panel sequencing that included the known and candidate genes involved in DSD (Table 3). Of 22 patients who were analyzed, five patients had the 46,XX karyotype, and 17 patients had the 46,XY karyotype. Three of the five tested 46,XX patients (60%) presented pathogenic variants (two patients) and VUS (one patient). Patient p9 (a 46,XX patient) presented compound heterozygous variants in the *RSPO1* gene, which were interpreted as class III variants. These variants included one c.286+1G>A (modifies the splice site between exon 4—intron 4, not noted in gnomAD, described in Clinvar [patient 18085567], predicted as pathogenic by *BayesDel_addAF*, *DANN*, *EIGEN*, *FATHMM-MKL*, *MutationTaster*, and *scSNV-Splicing*) and one c.484A>G (missense variant in exon 6, described in gnomAD as having a very low frequency [ƒ = 0.00000401]). Another 46,XX patient, p12, presented compound heterozygous variants in the *FGFR1* gene, one in c.914A>G, which was interpreted as pathogenic (a missense mutation in exon 8, a hot-spot region of the gene, classified as pathogenic by Uniprot, already described in several articles, not found in gnomAD, predicted as pathogenic by *BayesDel_addAF*, *DANN*, *DEOGEN2*, *EIGEN*, *FATHMM-MKL*, *LIST-S2*, *M-CAP*, *MVP*, *MutationTaster*, and *SIFT*) and one in c.2440A>C, which was interpreted as VUS (missense variant in exon 19, not found in gnomAD, predicted as pathogenic by the prediction platforms *BayesDel_addAF*, *DANN*, *DEOGEN2*, *EIGEN*, *FATHMM-MKL*, *LIST-S2*, *M-CAP*, *MutationAssessor*, *MutationTaster*, and *SIFT*). Patient p34 presented a heterozygous c.1075G>T variant in the *ATM* gene, which was interpreted as pathogenic (a null variant in exon 9, not found in gnomAD with pathogenic predictions in *BayesDel_addAF*, *DANN*, *EIGEN*, *FATHMM-MKL*, and *MutationTaster*); this gene has been associated with premature ovarian failure, as observed in patient p34.

Of the 17 46,XY patients tested by gene panel sequencing, 13 (76%) presented pathogenic variants (two patients), likely pathogenic variants (seven patients), and VUS (four patients). Patient p1 presented a heterozygous variant, c.437C>A, in the *WT1* gene, which was interpreted as VUS (a missense variant in exon 1, not described in gnomAD; 93% of missense variants of the WT1 gene are pathogenic, predicted pathogenic by *BayesDel_addAF*, *DANN*, *FATHMM-MKL*, *M-CAP*, *PrimateAI*, and *SIFT*). Patient p2 presented a heterozygous variant, c.5405-7G>A, in the *CHD7* gene, which was classified as likely pathogenic (an intronic variant, classified as pathogenic in Clinvar by several submissions and articles, not found in gnomAD).

Four 46,XY patients (p5, p8, p13, and p16) presented variants in the *AR* gene. In patient p5, the variant c.2071_2073del was classified as likely pathogenic (an in-frame deletion, not described in gnomAD, found in the hot-spot region of the *AR* gene, predicted as pathogenic by *phyloP*). Patient p8 presented two VUS variants, c.167T>A and c.170_171insGCAGCAGCA (not found in gnomAD, predicted as pathogenic by *phyloP*). In patient p13, the variant c.1097A>C was classified as VUS (missense variant in exon 1, not found in gnomAD, 96% of missense variants in the *AR* gene are pathogenic, prediction as pathogenic by the prediction platforms *BayesDel_addAF*, *DANN*, *FATHMM-MKL*, *LIST-S2*, *M-CAP*, *MVP* and *SIFT*). For patient p16, the variant in *AR* gene was interpreted as likely pathogenic (a missense variant in exon 6, in a known hot-spot mutation, not found in gnomAD and predictions of pathogenicity in: *BayesDel_addAF*, *DANN*, *FATHMM-MKL*, *LIST-S2*, *M-CAP*, *MVP*, *MutationTaster*, *PrimateAI*, and *SIFT*).

Patient p7 presented a pathogenic variant on the *NIPBL* gene (a null frameshift variant, not found in gnomAD, Clinvar classification as pathogenic, pathogenic prediction on *phyloP*). Patient p11 presented two variants in the *AMHR2* gene, one pathogenic (c.233-2A>G, null variant within splice site, not found in gnomAD, and predictions sustained the pathogenicity: *BayesDel_addAF*, *EIGEN*, *FATHMM-MKL*, *MutationTaster*, and *scSNV-Splicing*) and the other likely pathogenic (c.133delA, null frameshift variant, not found in gnomAD). Patient p15 had a likely pathogenic variant, c.512G>A, in the *NR0B1* gene (null nonsense variant, not found in gnomAD). Patient p17 presented a VUS variant, c.614G>A, in the *EMX2* gene (not found in gnomAD, pathogenicity predictions from: *BayesDel_addAF*, *DANN*, *DEOGEN2*, *EIGEN*, *FATHMM-MKL*, *LIST-S2*, *M-CAP*, *MVP*, *MutationTaster*, and *PrimateAI*). Patient p19 presented two variants in *CYP17A1*, one classified as likely pathogenic (c.1318C>T, a missense variant in exon 8, very small frequency on gnomAD, 97% of missense variants are pathogenic, the alternative variant was classified pathogenic by Uniprot, predictions for pathogenicity in: *BayesDel_addAF*, *DANN*, *DEOGEN2*, *EIGEN*, *FATHMM-MKL*, *LIST-S2*, *M-CAP*, *MVP*, *MutationAssessor*, *MutationTaster*, and *SIFT*) and the other VUS (c.1214A>G, a missense variant in exon 7, not found in gnomAD with predictions of pathogenicity in *DANN*, *DEOGEN2*, *EIGEN*, *FATHMM-MKL*, *LIST-S2*, *M-CAP*, *MutationAssessor*, *MutationTaster*, and *SIFT*).

Patient p25 presented the variant c.1214A>G in the *GNRHR* gene, which was noted as likely pathogenic (missense mutation in a functional domain of the gene, not found in gnomAD with predictions of pathogenicity in: *BayesDel_addAF*, *DANN*, *DEOGEN2*, *EIGEN*, *FATHMM-MKL*, *M-CAP*, *MVP*, *MutationAssessor*, *MutationTaster*, and *SIFT*). However, this gene is autosomal recessive, and the variant, heterozygous, was not sufficient to develop the pathology.

Patient p28 presented the variant c.698C>A in the GATA4 gene, which was classified as likely pathogenic (a missense variant in a functional domain of the gene, not found in gnomAD with predictions of pathogenicity in *BayesDel_addAF*, *DANN*, *DEOGEN2*, *EIGEN*, *FATHMM-MKL*, *M-CAP*, *MVP*, *MutationTaster*, *PrimateAI* and *SIFT*).

## 4. Discussion

In this study of 267 patients with DSD, of the 93 patients that completed all applicable tests, a diagnosis was obtained for 87 (94%) of the patients. A total of 174 patients did not pursue genetic testing and remained etiology unknown following karyotyping, *SRY* testing, and *CYP21A2* strip testing. Karyotype testing established the diagnosis in 15% of patients, most of whom presented abnormalities of the sex chromosomes. A value of basal 17-hydroxyprogesterone above the threshold of 2 ng/mL or of stimulated 17-hydroxyprogesterone above 10 ng/mL indicated a deficiency of 21-hydroxylase, and genetic testing of *CYP21A2* confirmed the diagnosis in 17 out of 39 patients (44% of *CYP21A2* tested patients).

SNP array analysis was particularly useful in DSD patients who presented associated signs (syndromic DSD). Of patients tested by SNP array, 20% received a final etiologic diagnosis, and almost all of them, with one exception, presented syndromic DSD (this percentage was similar between 46,XX and 46,XY DSD). Gene panel sequencing, which included DSD-associated genes (known and candidate) established or strongly suggested the etiologic diagnosis in 11 of 22 patients tested (50%). The androgen receptor gene mutations were most observed in the study group, but variants in some genes less commonly associated with DSD were also observed, such as those in the *GATA4* gene.

Regarding the percentage of chromosomal abnormalities observed in patients with DSD, a similar result (around 15%) has been found by other studies, and most of these abnormalities involve the sex chromosomes [18,19]. Thus, karyotype has a role not only in establishing the first step in a pathogenetic algorithm (in 46,XX or 46,XY DSD) but also in observing an etiology in a number of cases (in sex chromosome DSD). The karyotype also provides advantages for identifying mosaicisms, translocations, and X chromosome structural variants, which in some situations are not easily suggested using chromosomal microarray [13].

Concerning the CNVs observed in the study group, the percentage of pathogenic CNVs and uniparental disomy was 20%, similar to that found in other studies, which ranged from 15–20% [18,20,21,22,23]. One study found that chromosomal microarray is similar to classical cytogenetics when identifying Turner syndrome [24]. However, chromosomal microarray analysis has advantages over karyotype regarding the identification of cryptic Y chromosome material in patients with Turner syndrome [13]. For example, 1q21 deletion (involving *RBM8A* gene), identified in a 46,XX patient, is a described etiology in Mayer-Rokitansky-Küster-Hauser syndrome, which is present in some patients with this BP2-BP3 deletion [25].

Another patient was diagnosed with 21-hydroxylase deficiency due to homozygous deletion identified by SNP array testing. The presence of maternal uniparental disomy of 15 and uniparental disomy of chromosome 7 in three of the 35 patients studied by chromosomal microarray provided useful information, as the phenotype of these disorders (Prader-Willi or Russell-Silver syndromes) includes signs of DSD usually associated with other findings that are less obvious. Concerning the other pathogenic CNVs or VUS described in the study, a clear association was not found between genes included in these regions and DSD; these CNVs or genes have been more frequently associated with neurologic development than that of other systems or organs.

High-throughput sequencing was found to be the most effective in cases of DSD patients due to its high percentages of positive results. The findings of high-throughput sequencing indicated etiologic and pathogenetic mechanisms, which could be very important for designing optimal therapy and achieving a better prognosis. Of the study’s patients, 50% had a pathogenic or likely pathogenic variant responsible for their clinical development, and a similar percentage was observed in a large international cohort of DSD patients [26]. However, this percentage was higher than that observed in other studies on DSD patients (around 30%) [27,28,29,30,31,32]. This higher percentage may have been due to the fact that the patients tested by sequencing comprised those with a severe DSD phenotype [28]. A similar percentage of diagnosis was found for both 46,XY and 46,XX patients. The genes that presented pathogenic or likely pathogenic variants or VUS were *RSPO1*, *FGFR1*, *WT1*, *CHD7*, *AR*, *NIPBL*, *AMHR2*, *AR*, *EMX2*, *CYP17A1*, *NR0B1*, *GNRHR*, *GATA4*, and *ATM* genes. As in other studies, the frequency of *AR* gene mutations was higher than that of other genes [28,33,34,35].

The variant described in the *GATA4* gene, c.698C>A (p.Thr233Lys), fell within the N-terminal zinc finger region, one that was recently proved by van der Bergen to have a pathogenic effect in 46,XY DSD, unlike variants in other regions of this gene, which usually have a benign contribution [36]. Another pathogenic variant identified in the present study was in the *NIPBL* gene; this variant is associated with Cornelia de Lange syndrome. The genetic diagnosis for this patient was due to genital abnormalities seen and less to the dysmorphic and neuropsychiatric signs classically associated with this syndrome.

A limitation of the present study was the impossibility of evaluating the parents to establish the de novo or inherited characteristics of the unknown variants, as well as the inability to perform functional studies for these variants. However, an important contribution of this study was that it was the first in the authors’ country to perform a well-defined algorithm for genetic testing in DSD.

In conclusion, an evaluation following a genetic testing algorithm including karyotype and *SRY* gene testing, *CYP21A2* analysis, chromosomal analysis by microarray, and next-generation sequencing provided a diagnosis for 87 patients with a spectrum of diagnostic yield between 15 and 50%, depending on the technique. Additionally, new genetic variants not previously described in DSD patients were observed.

## Figures and Tables

**Figure 1 diagnostics-11-02107-f001:**
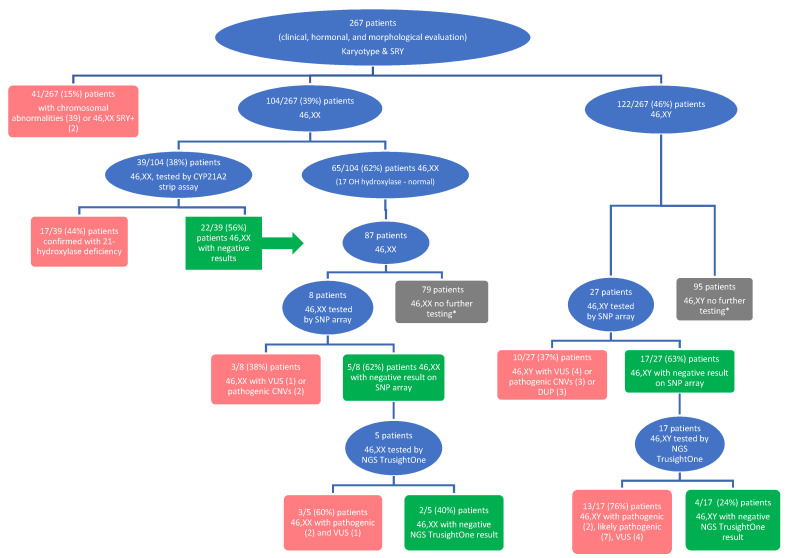
The investigation protocol for genetic testing in the study group. (* The patients were not further tested due to the absence of clinical monitoring after their first medical visit).

**Table 1 diagnostics-11-02107-t001:** Karyotypes observed in DSD patients.

Karyotype	Number of Patients, n
45,X	18
45,X/46,XY	2
45,X/46,XX	4
45,X/46,XX/47,XXX	3
46,Xi(Xp)	1
46,XX,dup(X)(q22q24)	1
47,XXY	3
46,XY/47,XXY	1
48,XXXY	1
48,XXYY	1
49,XXXXX	1
46,XY,dup(8)(q24.3)	1
46,XY,del(12)(q13)	1
46,XY/47,XY,+mar	1
46,XX	106
46,XY	122

**Table 2 diagnostics-11-02107-t002:** Pathogenic CNVs and VUS observed in patients analyzed by SNP array analysis.

Patient	Isolated/ Syndromic	Clinical Phenotype	Karyotype	CNV/DUP	Size	Main Genes Included	Interpretation
p3	Syndromic	Primary amenorrhea, clitoridian hypertrophy, short stature, intellectual disability	46,XX	arr[GRCh37]15q11.2(22753733_23226254)x3	472 kb	*NIPA1*, *NIPA2*, *CYFIP1*, *TUBGCP5*	VUS
p4	Syndromic	Bilateral cryptorchidism, intellectual disability, craniofacial dysmorphism	46,XY	UPD7	-	*-*	Pathogenic
p6	Syndromic	Enlarged phallus, urogenital sinus, partial posterior fusion of genital folds (Prader 3), global developmental delay, craniofacial dysmorphism	46,XY	arr[GRCh37]15q11.2(22652330_23226254)x3	573 kb	*NIPA1*, *NIPA2*, *CYFIP1*, *TUBGCP5*	VUS
p10	Syndromic	Cryptorchidism, micropenia, puberty delay, intellectual disability	46,XY	arr[GRCh37]16p11.2(29595483_30192561)x1	597 kb	*PRRT2*, *KIF22*, *ALDOA*, *TBX6*, *SEZ6L2*, *TMEM219*, *MAPK3*, *SPN*, *QPRT*, *MAZ*, *MVP*, and 11 other OMIM genes *(proximal 16p11.2 microdeletion syndrome)*	Pathogenic
p14	Syndromic	Bilateral cryptorchidism, short stature, obesity, deafness	46,XY	UPD15	-	*-*	Pathogenic
p18	Syndromic	Cryptorchidism, obesity, macrocrania, micromelia	46,XY	arr[GRCh37]Xp11.4(38230704_38246882)x1	16 kb	*OTC (exon 4)*	Pathogenic
p21	Syndromic	Micro-penis, bilateral, cryptorchidism, hypotonia, craniofacial dysmorphism	46,XY	UPD15	-	*-*	Pathogenic
p24	Syndromic	Scrotal hypospadias, cryptorchidism, intellectual disability, language delay, obesity	46,XY	arr[GRCh37]6p25.1(5256116_ 5391419)x1	135 kb	*FARS2*, *LYRM4*	VUS
p26	Syndromic	Micro-penis, cryptorchidism, craniofacial dysmorphism, intellectual disability	46,XY	arr[GRCh37]14q11.2(19401281_20420338)x3	1.01 Mb	*POTEG*	VUS
p30	Syndromic	Primary amenorrhea, uterus agenesia, forearm agenesia, ectromelia, epilepsy, intellectual disability	46,XX	arr[GRCh37]1q21.1(145394955_145755813)x1	360 kb	*RBM8A*, *PEX11B*, *POLR3GL*, *HJV*, *HEF2A*, and other OMIM genes (*BP2-BP3 microdeletion syndrome*)	Pathogenic
p31	Syndromic	Cryptorchidism, micro-penis intellectual disability, craniofacial dysmorphism	46,XY	arr[GRCh37]18p11.32p11.31 (13034_4390081)x1	4.3 Mb	*SMCHD1*, *LPIN2*, *TGIF1*, and other OMIM genes included in 18p11.3 region (*18p11.3 deletion syndrome*)	Pathogenic
p32	Isolated	Enlarged phallus, partial fusion of genital folds, Prader 2	46,XX	arr[GRCh37]6p21.33(32005904_32006896)x0	0.99 kb	*CYP21A2 (exons 1-3)*	Pathogenic
p35	Syndromic	Scrotal hypospadias, cryptorchidism, palatine cleft, skeletal dysplasia	46,XY	arr[GRCh37]Xp11.4 (41665315_41684603)x1	19 kb	*CASK* (*intron 2*)	VUS

CNV—copy number variants, UPD—uniparental disomy, VUS—variant of uncertain significance.

**Table 3 diagnostics-11-02107-t003:** Phenotypic and genotypic characterization of patients analyzed by sequencing.

Patient	Age	Gender	Gonads	Muller Derivatives	External genitalia	Puberty	FSH LH	Hormonal	Other	Karyotype	SRY	Variant (HGVS)	Zigosity	Variant Interpretation (ACMG Criteria)
p1	4	F	Bilateral testicular ectopia	Yes, uterus, Vagina	Female N	Tanner 1	Prepuberty level	N	craniofacial dysmorphism, diaphragmatic hernia, calos body hypoplasia	46,XY	+	*WT1*(NM_024426.5):	Heterozygous	VUS
												c.437C>A (p.Trp151Cys)		(PM2, PP2, PP3)
												MAF = 0		
p2	11	M	Bilateral cryptorchidism	No	Micropenis, cryptorchidism	Tanner 1	Prepuberty level	N	Craniofacial dysmorphism	46,XY	+	*CHD7* (NM_017780.3):	Heterozygous	Likely pathogenic
												c.5405-7G>A, MAF = 0		(PP5, PM2)
												rs398124321, MAF = 0		
p5	1	F	Bilateral cryptorchidism	No	Female N	Tanner 1	Prepuberty level	N	No	46,XY	+	*AR*(NM_000044.6):	Heterozygous	Likely pathogenic
												c.2071_2073del (p.Asp691del), MAF = 0		(PM1, PM2, PM4, PP3)
p7	1.6	M	Anorhidia	No	Micro-penis	Tanner 1	↑↑↑	N	craniofacial	46,XY	+	*NIPBL*(NM_133433.4):	Heterozygous	Pathogenic
									dysmorphism, GDD/ID			c.1808del (p.Lys603SerfsTer11)		(PVS1, PM2, PP3, PP5)
												rs727503767, MAF = 0		
p8	0.3	F	Bilateral testicular ectopia	No	Clitoridian hypertrophy, 2 orifices, right inguinal hernia	Tanner 1	Prepuberty level	N	Hypercalcemia	46,XY	+	*AR*(NM_000044):	Heterozygous	VUS
												c.167T>A (p.Leu56Gln)		(PM2, PP2)
												rs868709351, MAF = 0		
												*AR*(NM_000044):	Heterozygous	VUS
												c.170_171insGCAGCAGCA (p.L57insGlnGlnGln)		(PM2,PP3)
												rs3032358, MAF = 0		
p9	3.1	M	Scrotal testes	No	Penoscrotal hypospadias, micro-penis, bifidus scrotum	Tanner 1	Prepuberty level	T, DHT ↓	no	46,XX	-	*RSPO1*(NM_001242908.2):	Heterozygous	VUS
												c.286+1G>A		(PM2, PP3, PP5)
												rs1570099690, MAF = 0		
												*RSPO1*(NM_001242908.2):	Heterozygous	VUS
												c.484A>G (p.Lys162Glu)		(PM2)
												rs36043533, MAF = 0.0000464		
p11	0.3	F	Bilateral testicular ectopia	Yes, uterus, vagina	Penoclitoridian organ, one orifice at the base of the gland, labioscrotal folds posteriorly fusioned, not palpable gonads, Prader 4	Tanner 1	Prepuberty level	N	no	46,XY	+	*AMHR2*(NM_020547.3):	Heterozygous	Pathogenic
												c.233-2A>G, MAF = 0		(PVS1, PM2, PP3)
												*AMHR2*(NM_020547.3):	Heterozygous	Likely pathogenic
												c.133delA (p.Thr45GlnfsTer23), MAF = 0		(PVS1, PM2)
												*AMHR2*(NM_020547.3):	Heterozygous	VUS
												c.137G>T (p.Gly46Val)		(PM2, PP2, PP3)
p12	17	F	Not evidenced	Yes, hypoplasic uterus	Female N	Tanner 1	↑	N	no	46,XX	+	*FGFR1*(NM_001174067.1):	Heterozygous	pathogenic
												c.914A>G (p.Glu305Gly)		(PS3, PM1, PM2, PP2, PP3, PP5)
												rs727505369, MAF = 0		
												*FGFR1*(NM_001174067.1):	Heterozygous	VUS
												c.2440A>C (p.Thr814Pro)		(PM2, PP2, PP3)
												MAF = 0		
p13	22	M	Bilateral cryptorchidism	No	Masculine Prader 5	Tanner 5	N	N	no	46,XY	+	*AR*(NM_000044.6):	Heterozygous	VUS
												c.1097A>C (p.Asn366Thr)		(PM2, PP2, PP3)
												MAF = 0		
p15	0.1	M	Bilateral cryptorchidism	No	Micro-penis, hypospadias, one orifice at the base, scrotal folds, inguinal hernia	Tanner 1	Prepuberty level	N	no	46,XY	+	*NR0B1*(NM_000475.5):c.512G>A (p.Trp171Ter); MAF = 0	Heterozygous	Likely pathogenic (PVS1, PM2)
p16	1.2	M	Scrotal testes	No	Micro-penis, pensoscrotal hypospadias	Tanner 1	Prepuberty level	N	no	46,XY	+	*AR*(NM_000044.6): c.2415C>G	Heterozygous	Likely pathogenic
												(p.Phe805Leu)		(PM1, PM2, PP2, PP3)
												MAF = 0		
p17	4	M	Scrotal testes	No	Hypospadias	Tanner 1	Prepuberty level	N	Pierre Robin sequence,	46,XY	+	*EMX2*(NM_004098):	Heterozygous	VUS
									calos body hypoplasia, hydrocephaly, GDD/ID			c.614G>A(p.Arg205Gln) MAF = 0		(PM2, PP3)
p19	48	F	Bilateral cryptorchidism	no	Female, upper region of vagina without external orifice	Tanner 1	↑↑↑	Adrenal insuficiency	adrenal insuficiency,	46,XY	+	*CYP17A1*(NM_000102.4):	Heterozygous	Likely pathogenic
									arterial hypertension			c.1318C>T (p.Arg440Cys)		(PM2, PM5, PP2, PP3)
												rs868228603, MAF = 0.0000119		
												*CYP17A1*(NM_000102.4):c.1214A>G(p.Glu405Gly)	Heterozygous	VUS
												MAF = 0		(PM2, PP2, PP3)
p20	4	M	Bilateral cryptorchidism	No	Male, Prader 5	Tanner 1	Prepuberty level	Adrenal insuficiency	Adrenal insuficiency	46,XY	+	-	-	-
p22	15	M	Scrotal testes	No	Male N	Tanner 3, Ginecomastia	N	T↑ estradiol↑	no	46,XY	+	-	-	-
p23	40	F	Not evidenced	Yes, uterus, vagina	Female N	Tanner 3	N	N	no	46,XX	-	-	-	-
p25	1	M	Bilateral cryptorchidism	No	micro-penis	Tanner 1	↓	N	no	46,XY	+	*GNRHR*(NM_000406.3):	Heterozygous	Likely pathogenic
												c.236T>C (p.Leu79Pro)		(PM1, PM2, PP2, PP3)
												MAF = 0		
p27	14	M	Bilateral cryptorchidism	No	Micro-penis, penoscrotal hypospadias	Tanner 4, Ginecomastia	N	N	Obesity, short stature, astigmatism, hypermetropia	46,XY	+	-	-	
p28	6	M	Bilateral cryptorchidism	No	penoscrotal hypospadias	Tanner 1	Prepuberty level	N	Dysmorphism, aortic bicuspidia, ventricular septal defect, GDD/ID	46,XY	+	*GATA4*(NM_002052.5):	Heterozygous	Likely pathogenic (PM1, PM2, PP2, PP3)
												c.698C>A (p.Thr233Lys)		
												MAF = 0		
p29	0.8	M	Unilateral cryptorchidism	No	Penoclitoridian glans, one orifice at the base, labioscrotal folds, one gonad at the left fold, Prader 3	Tanner 1	Prepuberty level	N	no	46,XY	+	-	-	-
p33	20	F	Ovarian agenesia	Yes, uterus, vagina	Female N	Tanner 3	↑↑↑	N	no	46,XX	-	-	-	-
p34	32	F	Ovarian agenesia	Yes, uterus, vagina	Female N	Tanner 3	↑↑↑	N	no	46,XX	-	*ATM*(NM_000051.4):c.1075G>T (p.Glu359Ter)	Heterozygous	Pathogenic
												MAF = 0		(PVS1, PM2, PP3)

VUS—variant of unknown significance, MAF—minor allele frequency, GDD—global developmental delay, ID—intellectual disability, N—normal values, M—male, F—female.

## Data Availability

The data presented in this study are available on request from the corresponding author.

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
