# Peer review of "Molecular and Cytogenetic Analysis of Romanian Patients with Differences in Sex Development"

_diagnostics, 2021, doi:10.3390/diagnostics11112107_

Round 1

Reviewer 1 Report

The manuscript “Molecular and cytogenetic investigation in patients with differences in sex development” describes the diagnostic work-up followed in a single Clinical Center with the objective of obtaining an etiologic diagnosis in a series of patients with DSD (initially 267 patients). Authors designed an algorithm that begun by adding a karyotype and SRY gene detection to the clinical, hormonal and morphological evaluation. That first step seems correct and allowed the description of a group of chromosomal abnormalities (sex chromosomes and other) (39/267 = 14.6%) plus 2 patients 46,XX SRY positive. From then, we cannot follow the work-up in the 2nd and 3rd DSD groups according to the Chicago Consensus: the 46,XX and the 46,XY. They are pooled together and separated according to their serum 17-OH-progesterone: those with increased 17-OH-P are tested for CYP21A2 gene, while the rest (164 including both karyotypes) plus those tested normal for CYP21A2 (we neither know their karyotypes) (198 patients in total) follow a new step for which, authors loss a majority of patients (163). As a result, only 35 patients went through a CNV analysis that characterized 13/35 patients, and those remaining without a diagnosis went to a NGS panel analysis that allowed characterization of sequence variations in genes related to DSD or to Hypogonadotropic hypogonadism in 15/22 patients.

Unfortunately, this work-up does not bring a clear road to understand the diagnoses.

- 1) Authors should reorganize their algorithm to let readers know how many 46,XX and 46,XY are included in each step.

- 2) Authors should describe briefly the TrusightOne panel analysed: total number of genes included and how many among them the authors initially think that they are well recognized DSD causing genes.

- 3) Lines 52-53: CYP11A1 and CYP17A1 genes are lacking among steroid hormone biosynthesis disorders.

- 4) Line 55: In 46,XY disorders of hormone action ….

- 5) Table 3 lacks a title and cannot be fully read. It includes the 22 patients that were analysed by WES. Patients carrying a sequence variation in a gene related to DSD or to hypogonadotropic hypogonadism should receive a final diagnosis.

- 6) Abstract should be re-designed.

- 7) English language should be improved.  

Author Response

Dear Reviewer,

Thank you, a lot, for all the suggestions you made, I believe that these are very useful to have a better manuscript.

According to these suggestions, I made several changes. I change the text structure and I added more details.

- 1) Authors should reorganize their algorithm to let readers know how many 46,XX and 46,XY are included in each step.

I modified the algorithm steps.

- 2) Authors should describe briefly the TrusightOne panel analysed: total number of genes included and how many among them the authors initially think that they are well recognized DSD causing genes.

I added this information in text.

- 3) Lines 52-53: CYP11A1 and CYP17A1 genes are lacking among steroid hormone biosynthesis disorders.

I modified.

- 4) Line 55: In 46,XY disorders of hormone action ….

I changed, according to your suggestion.

- 5) Table 3 lacks a title and cannot be fully read. It includes the 22 patients that were analysed by WES. Patients carrying a sequence variation in a gene related to DSD or to hypogonadotropic hypogonadism should receive a final diagnosis.

I modified, here it is a lot of information, I hope that the information is seen.

- 6) Abstract should be re-designed.

I modified, according to modifications made in text

- 7) English language should be improved.  

The manuscript was submitted for proofreading for English.

Thank you a lot,

Kind regards,

Reviewer 2 Report

This a report on a study of 267 patients that met the authors’ criteria for DSD.

The purpose of the study is unclear. From the title, abstract and “aim” given at the end of Background [line98-99] the aim of the study was to catalog the aetiology of 267 patients that met the authors’ criteria for DSD. However, an aetiology for DSD was established for, at best, less than 40% of the patients. The first line of Discussion “In this study we observed the advantages and limitations of the various genetic tests proposed in the testing algorithm of patients with DSD.” This statement infers an assessment of the capabilities of each of the tests used in the study, however, all that are given are the results of each test – no assessment, no comparisons to alternative tests. The end of the Discussion concludes with “an evaluation following a genetic testing algorithm, …, allows the diagnosis for approximately 34% of patients” suggesting that the report is a quality improvement assessment of the process by which this group determines a diagnosis for DSD. The authors need to be clear from beginning to end what the purpose of the report is.

267 patients were evaluated and “Written informed consent was obtained from the parents of all the participants in the study.” However, Figure 1 appears to indicate that some proportion of 163 patients did not provide informed consent. The authors need to clarify this.

“2.3. Next generation sequencing” The genetic test described here is Whole Exome Sequencing (“genes associated with human pathology (also called the clinical exome)”). NGS is just the technique to complete whole exome sequencing; NGS is not a test on its own. The authors should refer to these as WES results, not NGS.

 “62 patients were tested for CYP21A2 and for 28 of them (21%) the diagnosis was confirmed”.  However, 28 of 62 is 45% and 28 of 267 is 10: the authors need to show were 21% comes from.

Unless the authors can explain otherwise, VOUS (variant of uncertain significance) is the same as VUS (variant of unknown significance) with VUS being the more accepted acronym.

Having a VUS does not complete a diagnosis; it just means that a variant was detected in the genes that were tested.

In Figure 1, the arrow from the right of “34 patients without diagnosis” should point to “198 patients without diagnosis”, not to “164 patients without diagnosis”.

In Figure 1, for consistency, “226 patients without diagnosis” should appear on the right of “41 patients with …” because all other “without diagnosis” type boxes are to the right of their “diagnosed” boxes.

Table 2 lists Patient as p3, p4, p6, etc. while Table 3 lists Patient as 1DSD, 2DSD, 3DSD, … 22DSD. This leaves open the possibility, for example, that p3 and 3DSD are the same patient? It is unclear whether the Patient label is even needed since no individual is referred to by their label. Perhaps the authors should omit the Patient column (if Tables 2 and 3 are to be included).

If this is a report on the capabilities of the diagnostic algorithm the authors used then Tables 2 and 3, and even Table 1, are unnecessary. Simple counts of how many are diagnosed and how many are not by each step of the algorithm would suffice (which is essentially Figure 1).

If this is a report on the capabilities of the diagnostic algorithm then the authors should assess the capabilities of the algorithm among those patients that completed the entire algorithm. Of the 267 patients that met criteria, 104 patients completed the entire algorithm of whom 97 were diagnosed or 93% (IF you accept that a VUS result is a diagnosis, which is questionable). So, the algorithm is working much better than the 34% or 14 to 45% the authors conclude with. The large drop in success is caused by the loss of 163 from the study. The question of interest is then why do 163 patients drop out when facing the genetic analyses? Is this caused by the cost the genetic tests? Is it caused by concerns about genetic testing and the possible effects on insurance or employment for the patient and family members? The authors should address this (and explain how the costs of such tests are covered in the Romanian health care system).

The text contains many errors in spelling (for example, “pourcentage” should be “percentage”) and expression (for example “progesterone was superior to 2 ng/ml” should be “progesterone was greater than 2 ng/ml”). The authors need to have the entire text carefully edited.

Author Response

Dear Reviewer,

Thank you, a lot, for all the suggestions you made, I believe that these are very useful to have a better manuscript.

According to these suggestions, I made several changes. I change the text structure and I added more details. Further I responded to the observations you made.

The purpose of the study is unclear. From the title, abstract and “aim” given at the end of Background [line98-99] the aim of the study was to catalog the aetiology of 267 patients that met the authors’ criteria for DSD. However, an aetiology for DSD was established for, at best, less than 40% of the patients. The first line of Discussion “In this study we observed the advantages and limitations of the various genetic tests proposed in the testing algorithm of patients with DSD.” This statement infers an assessment of the capabilities of each of the tests used in the study, however, all that are given are the results of each test – no assessment, no comparisons to alternative tests. The end of the Discussion concludes with “an evaluation following a genetic testing algorithm, …, allows the diagnosis for approximately 34% of patients” suggesting that the report is a quality improvement assessment of the process by which this group determines a diagnosis for DSD. The authors need to be clear from beginning to end what the purpose of the report is.

I reorganize the text according to your suggestions.

267 patients were evaluated and “Written informed consent was obtained from the parents of all the participants in the study.” However, Figure 1 appears to indicate that some proportion of 163 patients did not provide informed consent. The authors need to clarify this.

I changed, in text. Indeed, the limited financial funds did not permit us to finish the investigation for all patients. I explained more in text.

“2.3. Next generation sequencing” The genetic test described here is Whole Exome Sequencing (“genes associated with human pathology (also called the clinical exome)”). NGS is just the technique to complete whole exome sequencing; NGS is not a test on its own. The authors should refer to these as WES results, not NGS.

In changed in text.

“62 patients were tested for CYP21A2 and for 28 of them (21%) the diagnosis was confirmed”.  However, 28 of 62 is 45% and 28 of 267 is 10: the authors need to show were 21% comes from.

Here I made an error, I had a limited period of time to submit, now I saw, please excuse-me for this error. I changed in text.

Unless the authors can explain otherwise, VOUS (variant of uncertain significance) is the same as VUS (variant of unknown significance) with VUS being the more accepted acronym.

Now I changed with VUS for CNVs and single nucleotide variant, to be more simple and clear.

Having a VUS does not complete a diagnosis; it just means that a variant was detected in the genes that were tested.

Indeed VUS does not mean diagnosis, due to limited time that I had between the test, I went further with genetic testing in cases with negative results. I changed and described in text.

In Figure 1, the arrow from the right of “34 patients without diagnosis” should point to “198 patients without diagnosis”, not to “164 patients without diagnosis”.

In Figure 1, for consistency, “226 patients without diagnosis” should appear on the right of “41 patients with …” because all other “without diagnosis” type boxes are to the right of their “diagnosed” boxes.

I modified the entire schema of the investigation, considering your suggestions.

Table 2 lists Patient as p3, p4, p6, etc. while Table 3 lists Patient as 1DSD, 2DSD, 3DSD, … 22DSD. This leaves open the possibility, for example, that p3 and 3DSD are the same patient? It is unclear whether the Patient label is even needed since no individual is referred to by their label. Perhaps the authors should omit the Patient column (if Tables 2 and 3 are to be included).

I changed, using the same system of codes for these patients.

If this is a report on the capabilities of the diagnostic algorithm the authors used then Tables 2 and 3, and even Table 1, are unnecessary. Simple counts of how many are diagnosed and how many are not by each step of the algorithm would suffice (which is essentially Figure 1). If this is a report on the capabilities of the diagnostic algorithm then the authors should assess the capabilities of the algorithm among those patients that completed the entire algorithm. Of the 267 patients that met criteria, 104 patients completed the entire algorithm of whom 97 were diagnosed or 93% (IF you accept that a VUS result is a diagnosis, which is questionable). So, the algorithm is working much better than the 34% or 14 to 45% the authors conclude with. The large drop in success is caused by the loss of 163 from the study. The question of interest is then why do 163 patients drop out when facing the genetic analyses? Is this caused by the cost the genetic tests? Is it caused by concerns about genetic testing and the possible effects on insurance or employment for the patient and family members? The authors should address this (and explain how the costs of such tests are covered in the Romanian health care system).

I modified in text as much as I could, according to your observations.

The text contains many errors in spelling (for example, “pourcentage” should be “percentage”) and expression (for example “progesterone was superior to 2 ng/ml” should be “progesterone was greater than 2 ng/ml”). The authors need to have the entire text carefully edited.

The manuscript was submitted for proofreading for English.

Thank you a lot,

Kind regards,

Reviewer 3 Report

There are several issues with this report. The English requires considerable attention throughout the text. The spelling analysis in word will detect many of these errors.

For the presentation of results there are several improvements required –

A lot of work is required to improve the results. For the CNVs in table 2 please indicate the genes that are involved in the  duplications or deletions and their relationship to the phenotype. Most of these are syndromic cases so it needs to be stated/discussed if more than one gene is involved in the phenotype for each patient ie does one gene with a CNV explain all of the phenotype or are other genes involved.

There needs to be more detail in the results section for several of the patients that underwent NGS. For example, patient 1 has a complex syndromic phenotype and a variant in WT1 – does the variant in WT1 explain all of the phenotype or only the DSD? In these complex syndromic forms of DSD, are more than 1 gene involved? A variant causing DSD and another gene involved in the somatic anomalies. Was this found in the NGS data.

Very important – the authors must provide information on the Minor Allelic Frequency (MAF) of each variant that they report. This can be found here - https://gnomad.broadinstitute.org/

The authors should not indicate the global MAF but indicate the subpopulation showing the highest frequency for the variant. They should also indicate the rs number for each variant if known. Eg for patient 1 -- WT1 WT1(NM_024426.5): c.437C>A (p.Trp151Cys) – the identification is rs1267712523.

There also needs to be some sort of in silico analysis for each of the variants to support their pathogenicity.

There also needs to be more information on how variants were classified as VUS or pathogenic for each case. For example patient 2 has a CHD7 (NM_017780.3):

c.5405-7G>A variant. This within the splice region but not an essential splice site. Why was this classified as likely pathogenic whereas the WT1 variant is VUS? Patient 13 has 2 variants in CYP17A1 – one is classified as VUS and the other as likely pathogenic

The zygosity should be indicated for each variant.

The discussion could be improved by considering the literature for some of these genes/variants.

For example the GATA4 variant GATA4(NM_002052.5): c.698C>A (p.Thr233Lys) is very interesting as it falls within the N-terminal zinc finger and recently a study by van den Bergen JA, et al Mol Genet Genomic Med. 2020 Mar;8(3):e1095 indicates that pathogenic variants causing DSD appear to fall within this motif. Other variants also need to be considered as they do provide information of interest to the field.

Author Response

Dear Reviewer,

Thank you, a lot, for all the suggestions you made, I believe that these are very useful to have a better manuscript.

According to these suggestions, I made several changes. I change the text structure and I added more details. Further I responded to the observations you made.

A lot of work is required to improve the results. For the CNVs in table 2 please indicate the genes that are involved in the duplications or deletions and their relationship to the phenotype. Most of these are syndromic cases so it needs to be stated/discussed if more than one gene is involved in the phenotype for each patient ie does one gene with a CNV explain all of the phenotype or are other genes involved.

I modified the text according to your suggestions.

There needs to be more detail in the results section for several of the patients that underwent NGS. For example, patient 1 has a complex syndromic phenotype and a variant in WT1 – does the variant in WT1 explain all of the phenotype or only the DSD? In these complex syndromic forms of DSD, are more than 1 gene involved? A variant causing DSD and another gene involved in the somatic anomalies. Was this found in the NGS data.

I changed the text according to your suggestions.

Very important – the authors must provide information on the Minor Allelic Frequency (MAF) of each variant that they report. This can be found here - https://gnomad.broadinstitute.org/

The authors should not indicate the global MAF but indicate the subpopulation showing the highest frequency for the variant. They should also indicate the rs number for each variant if known. Eg for patient 1 -- WT1 WT1(NM_024426.5): c.437C>A (p.Trp151Cys) – the identification is rs1267712523.

I changed, considering your observations.

There also needs to be some sort of in silico analysis for each of the variants to support their pathogenicity.

I added more information about the variant interpretation.

There also needs to be more information on how variants were classified as VUS or pathogenic for each case. For example patient 2 has a CHD7 (NM_017780.3): c.5405-7G>A variant. This within the splice region but not an essential splice site. Why was this classified as likely pathogenic whereas the WT1 variant is VUS? Patient 13 has 2 variants in CYP17A1 – one is classified as VUS and the other as likely pathogenic

I added more information about the variant interpretation.

The zygosity should be indicated for each variant.

I indicated the zygosity in table.

The discussion could be improved by considering the literature for some of these genes/variants. For example the GATA4 variant GATA4(NM_002052.5): c.698C>A (p.Thr233Lys) is very interesting as it falls within the N-terminal zinc finger and recently a study by van den Bergen JA, et al Mol Genet Genomic Med. 2020 Mar;8(3):e1095 indicates that pathogenic variants causing DSD appear to fall within this motif. Other variants also need to be considered as they do provide information of interest to the field.

I added more data in discussions.

Thank you a lot,

Kind regards,

Round 2

Reviewer 2 Report

Authors are to be complimented on a more focused report. It is much improved. Given that the focus is now on the presentation of the cases, the following are suggestions to improve the presentation.

The Background should be rearranged to match the sequence of investigation as presented in Figure 1. Thus, lines 67 – 81 should be moved up to become the second paragraph.

[38] “… represent a milder effect …”

[54] “In some cases, DSD is observed as part of other developmental disorders (e.g., syndromic DSD)” should be moved to [41] to be the second sentence following “… sex chromosome DSD [5]).”

[55-56] “In other instances, DSD is itself not an endocrine disorder but rather a developmental one,…” As the start of the paragraph, should be written as: “DSD may not be an endocrine disorder but rather a developmental one, …”

[61-62] “However, rare etiology of these disorders could be caused by mutations of STAR …”  is better written as: “Genetic mutations found among the remainder include: STAR …”

[65-66] “The etiology of disorders of steroid hormone action in 46,XY patients is represented by AR, AMHR, or LHR abnormalities.” is better written as: “Disorders of steroid hormone action in 46,XY patients are most often due to gene mutations in AR, AMHR, or LHR abnormalities.” 

This does leave the reader wondering about disorders of steroid hormone action in 46,XX patients.

[106-108] “A total of 267 patients were evaluated with karyotype and SRY testing (using FISH – fluorescent in situ hybridization - or PCR – polymerase chain reaction- techniques; Fig. 1).” is better written as: “A total of 267 patients were evaluated (Fig 1.) with karyotype and SRY testing (using either fluorescent in situ hybridization (FISH) or polymerase chain reaction (PCR) techniques).”

[108-111] “The patients were also evaluated by imaging studies (ultrasound, pelvic MRI), and for some, exploratory laparotomy, gonadal biopsy, and other investigations were also performed depending on clinical context.” Is better written as: “Patients were evaluated by imaging studies (ultrasound, pelvic MRI), with additional studies (such as exploratory laparotomy and gonadal biopsy) performed depending on clinical context.”

[117] “A total of sixty-two 46,XX patients were genetically tested …” is better written as: “Sixty-two 46,XX patients were genetically tested …”

[119-120] “For patients for whom the testing of the karyotype + SRY and the 21-hydroxylase deficiency did not establish a diagnosis, the SNP array …” is better written as: “For patients for whom karyotype, SRY and 21-hydroxylase deficiency testing did not establish a diagnosis, the SNP array …”

[121-126] “However, the need for supplementary biological samples or further clinical data from follow-up investigations was a reason to exclude a number of patients from further genetic testing. The limited financial funds for genomic testing provided by our healthcare system, such as SNP array or high throughput sequencing, leads us to the decision to perform these tests only for the patients who continued the follow-up for their disorders.” Not clear what is meant here.  Is the following correct: “Some patients declined follow-up investigations precluding obtaining further clinical data or biological samples. Therefore, given limited financial funds for genomic testing (SNP array or high throughput sequencing) provided by our healthcare system, it was decided to perform these tests only for the patients who continued the follow-up for their disorders.”

Are there any clinical, hormonal or morphological differences between the group of “no further testing” patients and those that did do further testing.  For example, did those with “no further testing” have milder phenotypes than those that did do testing? Did they have different average Prader scores? (Separate 46,XX and 46,XY for this.) Did they have phenotypes that were easily corrected by surgery so that their parents were no longer as anxious about their child? It is hard to believe that parents of a child with severe DSD would refuse follow-up. The authors should provide some more details about this no follow-up group, even if it is just to say that they had no clinical, hormonal or morphological differences from the tested group.

[126-129] “SNP array testing was indicated for 35 patients (Fig. 1), and a total of 22 patients were evaluated with gene panel sequencing (TruSight One panel, Illumina) after receiving a negative result from SNP array testing (Fig. 1).” is better written as: “SNP array testing was completed for 35 patients (Fig. 1), of whom 22 patients had negative results and were consequently evaluated with gene panel sequencing (TruSight One panel, Illumina) (Fig. 1).”

[133] “… parents of all participants in the study.” should be “… parents of all patients in the study.”  to be consistent with the remainder of the report.

[152-153] “This included known genes or candidate genes associated with the clinical phenotype for DSD (around 150 DSD genes were identified).” is better written as: “This included around 150 genes or candidate genes associated with the clinical phenotype for DSD.”

[Figure 1]

If the authors are going to provide a figure in colour then they should consider using colour to mean something. For example, boxes where a diagnosis is reached can be in one colour, while boxes for negative results (requiring further testing) can be in another colour. Perhaps the two boxes with “no further testing” can be in grey. Perhaps those boxes that show neither positive or negative results, for example, [76 patients 46,XX], [8 patients 46,XX tested by SNP array] can be in another colour and/or have a different shape (e.g., ovals)

The text in the boxes should be consistent, for example, [41/276 (15%) patients with chromosomal …], [104/267 (39%) patients 46,XX], [122/267 (46%) patients 46,XY].   It would also help to size the boxes so that first lines all appear like “41/276 (15%) patients”, “104/267 (39%) patients”, thus:

[ 41/276 (15%) patients

with chromosomal abnormalities (39)

or 46,XX SRY+ (2)  ]

[  104/267 (39%) patients

46,XX  ]

And for the top box

267 patients

(clinical, hormonal, …)

Karyotype & SRY

 Note: in that first box it should be “or 46,XX” not “and 46,XX”

“negative result – SNP array” and “ negative results SNP array” should both read “negative result on SNP array”

“(17 OH hydroxylase – N)” should be either “17 OH hydroxylase – negative”,  “17 OH hydroxylase – normal”  or  “17 OH hydroxylase ≤ 2 ng/ml”

In the caption for Figure 1, the font for “(*the patients were not further tested due to the absence of clinical monitoring after their first medical visit)” is too small.

[172-174] “In 28 (45%) of the 62 patients, the diagnosis was confirmed (homozygous or compound heterozygous), and 27 of these patients were heterozygous (43%).” is better written as: “In 28 (45%) of the 62 patients, the diagnosis was confirmed (1 homozygous, 27 compound heterozygous).”

[175-182] Move the (Table 2) reference to or add a (Table 2) reference after “… using chromosomal microarray.” on line [177]

[183] Should the “(37%)” come after the “10”? It is not clear what the 37% is referring to.

[189] “… cryptorchidism is not known.”

[220] “… SNP array testing were …”

[227 and elsewhere] “predicted as pathogenic by multiple prediction platforms”  Please list which platforms gave the predictions of pathogenicity rather than giving a vague reference to “multiple platforms”, for example, “predicted as pathogenic by X, Y and Z”.  Similarly [257-258 & 265-266 & 270-271 & 275 & 279] ‘’ several predictions of pathogenicity”, [260-261 & 267] “pathogenic predictions on several prediction platforms”, [262-263] “several predictions sustained the pathogenicity” should all list the platforms: “prediction as pathogenic by X, Y and Z”

[243] “93% of missense variants of the WT1 gene are pathogenic”

[250-251] “predicted as pathogenic by phyloP”

[252-253] “(not found in gnomAD, not predicted as pathogenic)”

[254-255] “96% of missense variants in the AR gene are pathogenic, prediction as pathogenic by …”

[269-270] “97% of missense variants are pathogenic, …”

[286-287] “In this study, the diagnostic efficiency of various genetic tests proposed in the testing algorithm of DSD patients was observed.”  The report would be better served if the Discussion began with an overall summary for your DSD protocol as a whole. For example: “In this study of 267 patients with DSD, of the 104 patients that completed all applicable tests a diagnosis was obtained for 98 (94%) of the patients. A total of 163 patients did not pursue genetic testing and remained etiology unknown following karyotyping, SRY testing and CYP21A2 strip testing. Karyotype testing established …”

[357-358] “Additionally, new genetic variants not previously described in DSD patients were observed.” How many new variants were observed?

Author Response

Dear Reviewer,

Thank you a lot for all your valuable mentions and suggestions, I appreciate a lot your availability and help in improving this paper. I modified the text according to all your suggestions. 

Kind regards

Reviewer 3 Report

My comments have been addressed

Author Response

Dear Reviewer,

Thank you a lot for all your valuable suggestions.

I made the English proofreading. 

Kind regards